# Steric and Electronic Effects in N-Heterocyclic Carbene Gold(III) Complexes: An Experimental and Computational Study

**DOI:** 10.3390/molecules27238289

**Published:** 2022-11-28

**Authors:** Miguel A. Rosero-Mafla, Jhon Zapata-Rivera, M. Concepción Gimeno, Renso Visbal

**Affiliations:** 1Departamento de Química, Facultad de Ciencias Naturales y Exactas, Universidad del Valle, A.A. 25360, Cali 760042, Colombia; 2Departamento de Química, Facultad de Ciencias, Universidad de los Andes, Cra 1 No 18A—12, Bogotá 111711, Colombia; 3Departamento de Química Inorgánica, Instituto de Síntesis Química y Catálisis Homogénea (ISQCH), CSIC-Universidad de Zaragoza, 50009 Zaragoza, Spain; 4Centro de Excelencia en Nuevos Materiales (CENM), Universidad del Valle, A.A. 25360, Cali 760031, Colombia

**Keywords:** Au(III), NHC, acridine derivatives, TD-DFT, pentafluorophenyl group

## Abstract

A series of neutral acridine-based gold(III)-NHC complexes containing the pentafluorophenyl (–C_6_F_5_) group were synthesized. All of the complexes were fully characterized by analytical techniques. The square planar geometry around the gold center was confirmed by X-ray diffraction analysis for complexes **1** (Trichloro [1-methyl-3-(9-acridine)imidazol-2-ylidene]gold(III)) and **2** (Chloro-bis(pentafluorophenyl)[1-methyl-3-(9-acridine)imidazol-2-ylidene]gold(III)). In both cases, the acridine rings play a key role in the crystal packing of the solid structures by mean of π–π stacking interactions, with centroid–centroid and interplanar distances being similar to those found in other previously reported acridine-based Au(I)-NHC complexes. A different reactivity when using a bulkier N-heterocyclic carbene ligand such as 1,3-bis-(2,6-diisopropylphenyl)-2-imidazolidinylidene (SIPr) was observed. While the use of the acridine-based NHC ligand led to the expected organometallic gold(III) species, the steric hindrance of the bulky SIPr ligand led to the formation of the corresponding imidazolinium cation stabilized by the tetrakis(pentafluorophenyl)aurate(III) [Au(C_6_F_5_)_4_]^−^ anion. Computational experiments were carried out in order to figure out the ground state electronic structure and the binding formation energy of the complexes and, therefore, to explain the observed reactivity.

## 1. Introduction

Since the first isolation one decade ago, N-heterocyclic carbenes (NHCs) have been treated as novel ligands in multiple investigations [1]. However, nowadays NHCs are among the most useful ligands, being used in a great variety of applications. There are plenty of reviews covering different areas in which NHCs play a key role [2,3,4,5]. Medicinal chemistry is still one of the principal research areas that takes advantage of the versatility of such ligands to explore new alternatives for the treatment of cancer [6,7,8]. In 2012, our research group reported the synthesis and characterization of several silver(I)- and gold(I)-NHC complexes that displayed blue–green emissions thanks to the presence of an acridine-based chromophore [9]. Later, in an extension of that work, various gold(I) complexes incorporating bioactive molecules and acridine-based NHC ligands were reported to be good antitumoral agents against two different cancer cell lines, A545 and MiaPaca2 (Figure 1) [10].

One of the major limitations observed for acridine-based M-NHC complexes is their poor solubility in moderate polar solvents. In general, the incorporation of bioactive molecules such as 2-mercaptopyridine or thio-β-D-glucose tetraacetate improves their solubility and therefore their biodistribution in the cell interior [10,11]. Fluorine-containing groups are known for enhancing polarity when introduced into non-polar molecules. Trifluorophenyl (-CF_3_) and pentafluorophenyl (-C_6_F_5_) groups are particularly well-known in this regard [12,13,14,15]. This pentafluorophenyl fragment has also been used to prepare many group 11 transition-metal complexes [16,17,18,19,20,21]. Its great sigma-donating properties and aromaticity help to stabilize metal ions, which in principle display some instability. For example, gold(III) is typically reactive/unstable under physiological conditions induced by intracellular redox reactions [22]. Taking this into account, the incorporation of the pentafluorophenyl group together with the presence of an appropriate NHC ligand could improve the stability of gold(III) species in physiological conditions for biological purposes. In fact, there are several works reporting on the synthesis and characterization of stable C_6_F_5_-containing gold(III)–carbene complexes (Figure 2) [23,24,25,26,27,28,29,30].

Most of these gold(III) complexes have been proven to be stable enough to be used as precursors in different catalytic reactions and also as anticancer agents against several cancer cell lines [22]. The aim of this work was to synthesize and characterize acridine-base gold(III)-NHC complexes and to explore the influence on the structural and electronic properties of the pentafluorophenyl group.

## 2. Results and Discussion

### 2.1. Synthesis of the Au(III) Complexes with NHC Ligands Derived from Acridine

The 1-methyl-3-(9-chloroacridine)imidazolium chloride ([IMeAcr-H]Cl) salt used as a precursor of the N-heterocyclic carbene ligand was obtained according to the procedure reported by Gimeno and co-workers [9] and then reacted with an equivalent amount of the corresponding gold(III) derivative, ([AuCl_3_(tht)], [Au(μ-Cl)(C_6_F_5_)_2_]_2_, or [Au(C_6_F_5_)_3_(tht)]), to generate the gold(III)-NHC complexes **1**–**3**, respectively (Figure 1).

This could be explained by the fact that the formation of the NHC ligand favored by the addition of the mild base K_2_CO_3_ is able to displace the labile group (tht) to afford complexes **1** and **3**, or to promote the dimer cleavage to form complex **2**. However, very recently, Nolan and co-workers reported an interesting mechanistic pathway in which the aurate(I) anion and a mild base, such as K_2_CO_3_ or NEt_3_, play a key role in the formation of the final [AuCl(NHC)] complex [31]. In this sense, and following the same hypothesis, a plausible reaction pathway is proposed for the formation of complexes **1**–**3**. Similarly, as observed for those gold(I) complexes previously reported [31], all of the intermediates (**Int1**–**Int3**) were detected by ^1^H NMR spectroscopy in the crude reaction after just 10 min of the reaction (See Appendix A). This is also associated with the higher stability of the intermediates compared to the corresponding precursors imidazolium chloride salts (Figure 2) [31].

Because of the small size of the methyl group on the imidazolium salt, the approach of the aurate(III) anion could not be so sterically hindered to promote the formation of a concerted bond-making/bond-breaking situation leading to the corresponding Au(III)-NHC complexes (**1**, **2** or **3**) and the spontaneous precipitation of KCl and KHCO_3_ (Figure 2) [31].

The influence of the incorporation of the pentafluorophenyl group in the acridine-based NHC-gold complexes was initially evidenced by better solubility, even in less polar solvents. They displayed the following tendency in solubility: [AuCl_3_(NHC)] (**1**) < [AuCl(C_6_F_5_)_2_(NHC)] (**2**) < [Au(C_6_F_5_)_3_(NHC)] (**3**). For example, complex **1** only displayed moderate solubility in polar solvents such as DMSO or methanol, while complexes **2** and **3** presented excellent solubility properties in solvents such as acetone, acetonitrile, dichloromethane and even chloroform (75 mg/mL). As expected, all the complexes obtained shared some ^1^H NMR spectroscopy features due to the hydrogen signals assigned to the acridine-based NHC ligand. The absence of a singlet at 9.95 ppm corresponding to the NCHN imidazolium proton confirms the formation of the Au(III)-NHC complexes [9]. Although ^13^C-{^1^H} NMR spectroscopy is normally used to characterize NHC-metal complexes, the C2 carbenic carbon could not be identified, even when using two-dimensional NMR spectroscopy. This may be indicative of a rapid relaxation process of the carbon nuclei. In addition, the differences among their chemical shifts and their multiplicity depend on both the deuterated solvent used and the gold(III) center, which for the obtained complexes had a different extent of functionalization due to the presence of pentafluorophenyl groups [27,29]. Nevertheless, in the mass spectra, the molecular peak [M-2Cl + H]^+^ appeared at *m*/*z* = 492.0562, and [M-Cl]^+^ appeared at *m*/*z* 790.0523 for complexes **1** and **2**, respectively. For complex **3,** the peak observed at *m*/*z* = 958.0623 was assigned to [M + H]^+^ (See Appendix A).

It is noteworthy that the dimer cleavage promoted by the NHC ligand to form complex **2** only gave the cis isomer (Figure 1). This could be associated with the cis conformation of the dimer precursor [Au(μ-Cl)(C_6_F_5_)_2_]_2_ and also due to the steric hindrance imposed by the carbene ligand (Figure 1). Consequently, both gold(III) complexes (**2** and **3**) displayed two sets of signals in the ^19^F NMR spectra corresponding to the ortho-, meta- and para-fluorine atoms in the –C_6_F_5_ group ranging from −120.90 to −164.70 ppm, similar to those reported in the literature [27]. According to the electronic spectra solved herein, all of the gold(III) complexes displayed structured bands with two maxima at 250 and 360 nm (see Appendix A). These bands can be assigned to π→π*- or n→π*-type transitions within the acridine chromophore, which are in good agreement with the spectra previously obtained for the acridine-based NHC-gold(I) complexes and other reported data for acridine-based derivatives [32]. It is noteworthy that none of the complexes displayed photoluminescent properties. The high electrophilicity of the gold(III) ion could be associated with the luminescence quenching observed for these species, which can promote d–d transitions that are energetically close to emissive IL or MLCT states [3]. This fact makes these Au(III) derivatives inappropriate for use as biomarkers in living cell imaging agents.

### 2.2. X-ray Structure Analysis

Single crystals of gold(III) complexes **1** and **2** were obtained and have been analyzed by X-ray diffraction. Their solid structures are depicted in Figure 3 and Figure 4. The crystal structure of complex **1** was monoclinic and contained two molecules per asymmetric unit. As expected for gold(III) complexes, the geometry around the metal center was square planar. On the other hand, the solid structure of complex **2** was triclinic and displays one molecule per asymmetric unit and was crystallized as an acetone adduct (Figure 4) with similar bond angles associated with a square planar geometry. Both complexes had the acridine-based NHC ligand in their structures where the carbenic carbon binds the Au(III) metal atom, in which the disposition of the acridine and imidazole rings was almost perpendicular (89.82°) for one of the molecules in the asymmetric unit of complex **1**, whereas the torsion angle for complex **2** was found to be 65.78° (see Table 1).

Similar to other gold(I) complexes containing the acridine-based NHC ligand previously reported [26], a significant difference was found in the carbene–gold bond distances Au(1)–C(1) of 2.007(6) and 2.047(2) Å for complex **1** and **2**, respectively, after swapping a chloride ligand for a pentafluorophenyl group. This is in good agreement with the great trans-influence associated with the C_6_F_5_- fragment as compared to the chloride ligand. This is also seen when comparing Au1-Cl1 bond distances in complexes **1** and **2** (2.3198(15) vs. 2.3323(6)), in which the C_6_F_5_- anion seems to be a greater trans-influence group than even the acridine-based NHC ligand.

As expected, the presence of the acridine moiety in both complexes promoted the formation of π···π stacking interactions between the acridine rings of different molecules [9,33]. Calculation of planes and centroids allowed the determination of a py-py contact between the two aromatic rings with interplanar and centroid–centroid distances of 3.517 and 3.583 and 3.504 and 3.701 Å, for complexes **1** and **2**, respectively (Figure 5 and Figure 6). The centroid–centroid distances found are in good agreement with those obtained for other metal-based analogs containing an acridine moiety ranging from 3.585 to 3.820 Å [9,34,35]. Parallel disposition is normally associated with a displacement angle between the acridine rings. In this case, complex **2** showed a displacement angle of 18.77°, while that of complex **1** was only 11.10°, the smallest value found for the acridine-based gold-NHC complex. Although complex **3** was identified by NMR spectroscopy and mass spectrometry, no single crystals suitable for X-ray diffraction analysis were obtained.

### 2.3. Synthesis of the 1,3-Bis-(2,6-diisopropylphenyl)imidazolinium tetrakis(pentafluorophenyl)aurate(III) Salt

To explore the reactivity of the precursor [Au(C_6_F_5_)_3_(tht)] against a different NHC ligand, we prepared a complex with the general formula [Au(C_6_F_5_)_3_(NHC)] using the bulkier ligand 1,3-Bis-(2,6-diisopropylphenyl)imidazolidine-2-ylidene (SIPr), which is one of the bulkiest ligands among the NHCs commonly used in different applications [36]. To obtain the desired complex, [Au(C_6_F_5_)_3_(SIPr)], and the acridine-based Au(III)-NHC complexes, a reaction between the imidazolinium salt [SIPr-H]Cl and [Au(C_6_F_5_)_3_(tht)] was induced in the presence of K_2_CO_3_. However, the presence of the bulky NHC ligand (SIPr) possibly prevented the formation of the desired gold-NHC complex due to the steric hindrance (Figure 3). In fact, the direct reaction of the free SIPr with [Au(C_6_F_5_)_3_(tht)] did not afford the complex [Au(C_6_F_5_)_3_(SIPr)] **4***. The limitations in the formation of the organometallic complex (**4***) were evidenced by the gradual increase in the purple color in the reaction mixture, which is characteristic of the formation of gold nanoparticles and the final formation of species **4**. As reported by Nevado and co-workers, this could be rationalized as the result of three steps. First, the detection of decafluorobiphenyl (C_6_F_5_–C_6_F_5_) in the initial crude reaction suggests that a cross-coupling reaction can be achieved via reductive elimination of an increasingly small amount of the tris(pentafluorophenyl)gold(III) intermediate promoted by the presence of K_2_CO_3_ to afford [SIPr-H][Au(C_6_F_5_)Cl] [37]. Subsequently, the excess of K_2_CO_3_ remaining in the crude reaction could also lead to the formation of gold nanoparticles and scrambling of ligands (C_6_F_5_^−^ and unknown side products) through a disproportion reaction of the chloridopentafluorophenylaurate(I) anion [37,38]. Finally, the unreacted intermediate ([SIPr-H][Au(C_6_F_5_)Cl]) remaining in the reaction medium is able to rapidly trap a C_6_F_5_^−^ fragment to form the tetrakis(pentafluorophenyl)aurate(III) [Au(C_6_F_5_)_4_]^−^ anion, which is stabilized by the imidazolinium cation to afford **4** (Figure 3).

The ^1^H NMR spectrum of gold–imidazolinium salt **4** showed a singlet at 7.72 ppm characteristic of the N(N)C-H proton of the N-heterocycle, which was strongly upfielded with respect to that observed for the precursor [SIPr-H]Cl (9.55 ppm) and also the intermediate, which appeared at 8.05 ppm (see Appendix A). The other signals did not show significant changes relative to the chemical shifts in the ^1^H NMR spectra. The [Au(C_6_F_5_)_4_]^−^ counteranion showed two multiplets at −122.36 and −162.53 ppm and a triplet at −159.50 ppm in the ^19^F NMR spectrum with ^3^*J_F-F_* = 19.8 Hz.

Single crystals of gold–imidazolinium salt **4** suitable for X-ray diffraction analysis were obtained, the solid structure of which is depicted in Figure 7. Although there are a few examples of salts (including imidazolium) being stabilized by gold(III) anions [39,40,41,42,43,44], to the best of our knowledge, this is the first solid-state structure of an imidazolinium cation stabilized by an aurate(III) anion reported in the literature.

As observed, the bond angles were close to those found for gold(III) centers with square planar geometry. Compound **4** crystallized in the monoclinic space group P2(1)/c, with Au-C bond lengths and angles similar to those found for other previously reported tetrakis(pentafluorophenyl)gold(III) derivatives [39]. In the crystal packing (Figure 8), the imidazolinium cations were arranged in coupled rows along the b axis, as if related by a center of symmetry, with the anionic [Au(C_6_F_5_)_4_]^−^ units filling the gaps between the rows.

Several attempts to use the [IPr-H]Cl salt under the same reaction conditions were unsuccessful due to the presence of a mixture of products that were very difficult to separate by means of standard separation techniques.

### 2.4. Computational Studies

To understand the reactivity observed, calculations were carried out based on the framework of the density functional theory (DFT) [45]. Geometry optimization calculations were performed on the reactants, products and intermediates associated with the formation of complexes **3** and **4**, allowing us to obtain more stable structures and the corresponding reaction energies (see SI for computational details). The intermediates’ and products’ relative electronic energies were estimated as ∆*E*_int/prod_ = *E*_intermediate/product_ − *E*_NHC_ − *E*_Au-complex_. As shown in Table 2, all energies indicated a thermodynamically favorable intermediate (**Int4**) and product formation ∆*E*_int_ < −47 kcal·mol^−1^ and ∆*E*_prod_ < −90 kcal·mol^−1^, respectively. It is worth mentioning that thermal and entropic corrections of the electronic energies did not have a significant impact on the relative energies (Appendix A). The amplitudes of these formation energies are characteristic of comparable gold-NHC complexes reported in the literature [31,46]. Because the reaction for the formation of complex **4** differed from that occurring for the formation of complex **3**, we also considered the formation of a hypothetical complex [Au(C_6_F_5_)_3_(SIPr)] (**4***), Figure 3. The lower ∆*E*_prod_ of **4*** (−114.6 kcal·mol^−1^) suggests its formation over that of the aurate(III) anion in salt **4**. However, the alternative formation of **Int5** after reductive elimination of the [Au(C_6_F_5_)_3_Cl]^−^ complex at −69.2 kcal·mol^−1^, as well as its lesser steric hindrance, likely leads to a lower barrier to reaching **4**. Hence, we expect the detection of **4** as the main product of the reaction mixture, as it was observed in our experiments.

DFT calculations also enabled us to roughly estimate the electronic structure of products **1**–**4**, based on the one-electron diagram of the σ interaction between the NHC and gold moieties. As depicted in Figure 9 and Appendix A, the $\sigma^{*}$ orbital of complexes **1**–**4** had a larger contribution of the Au 5d_x^2^−y^2^_ atomic orbital. The other four 5d orbitals of the gold atom remained double-occupied. That electronic distribution is in line with a dominant d^8^ configuration in the metal center, i.e., a gold(III) complex. This inference is also supported by the sign and large amplitude of the Mülliken population on the gold ranging from 0.4704 to 0.7296. In addition, the electronic structure of the NHC and Au(III) fragments was evaluated separately.

Based on the analysis of the σ molecular orbitals of these fragments, the formation of complexes **1**–**4** could be understood as a result of the p-d interaction between the electron-donating NHC moiety (C_6_F_5_- for **4**) and the electron-withdrawing gold complex, respectively (Figure 9 and Appendix A). In all cases, the double-occupied p orbital was mainly located in the carbenic carbon, whereas the unoccupied orbit was mostly of a d nature in the Au fragment.

To describe the nature of the main transitions associated with the bands of the UV-Vis spectra, we also simulated the absorption spectrum of complex **3** using TD-DFT calculations, including the 50th low-lying excited states (Appendix A). Two bands were found around 337 and 223 nm associated mostly with the 1st and 13th excited states, respectively, in agreement with those bands at 360 and 250 nm observed in the experiments. Table 3 shows the dominant electronic transitions of those states with a larger oscillator strength at each band. The analysis of the excited states’ electronic structure revealed three types of transitions: π_acr_-π_acr_, π_pfp_-π_acr_ and π_acr_-d, that according to their nature, have been classified as intraligand (IL), ligand-to-ligand (LL), and ligand-to-metal charge transfer (LMCT) transitions, respectively. The weight of the dominant transitions reported in Table 3 shows that the higher intensity band was of an IL nature, as anticipated from the experiments. However, non-negligible LMCT transitions appeared in the same UV-Vis region as a consequence of the strong interaction between the NHC and the Au(III) center. It is worth mentioning that d-d transitions with small oscillator strength have been observed at higher energy bands, and these transitions could be associated with the fact that complex **3** exhibits a non-radiative decay [3]. Comparable behavior to **3** was observed in complexes **2** and **1**.

## 3. Materials and Methods

The starting materials, [AuCl_3_(tht)] [47], [Au(μ-Cl)(C_6_F_5_)_2_]_2_ [48], and [Au(C_6_F_5_)_3_(tht)] [49], and the starting imidazolium salt, 1-(9-acridine)-3-methylimidazolium chloride [IMeAcr-H]Cl [9], were prepared according to published procedures. All other starting materials and solvents were purchased from commercial suppliers and used as received unless otherwise stated.

The general procedure for the synthesis of complexes **1**, **2** and **3** was as follows. To a suspension of the imidazolium salt [IMeAcr-H]Cl (147.9 mg, 0.5 mmol) in dried dichloromethane (20 mL), the corresponding gold(III) precursor, [AuCl_3_(tht)] (0.5 mmol), [AuCl(C_6_F_5_)_2_]_2_ (0.25 mmol) or [Au(C_6_F_5_)_3_(tht)] (0.5 mmol), was added. An excess of K_2_CO_3_ (10 mmol) was added to the mixture, which was then stirred for 2 h and filtered over celite at room temperature. The yellow filtrate was evaporated until the minimum volume (c.a. 2 mL) under vacuum was reached, and after addition of hexane (15 mL), a yellow precipitate was obtained corresponding to complexes **1**, **2** and **3**.

### 3.1. General Measurements and Analysis Instrumentation

Mass spectra were recorded on a Bruker Esquire 3000 Plus, with the electrospray (ESI) technique. UV−Vis spectra were recorded with 1 cm quartz cells on an Evolution 600 spectrophotometer (Thermo Electron Scientific Instrument LCC, Madison, WI, USA). ^1^H, ^13^C{^1^H} and ^19^F NMR (CFCl_3_ used as standard), including 2D experiments, were recorded at room temperature on a Bruker Avance 400 spectrometer (Bruker, Billerica, MA, USA) (^1^H, 400 MHz, ^13^C, 100.6 MHz, ^19^F, 376.5 MHz) or on a Bruker Avance II 300 ((Bruker, Billerica, MA, USA) (^1^H, 300 MHz; ^13^C, 75.5 MHz; ^19^F, 282.3 MHz) with chemical shifts (δ, ppm) reported relative to the solvent peaks of the deuterated solvent [50].

### 3.2. Crystallographic Data

Crystal structure determinations were carried out as follows. Crystals were mounted in inert oil on glass fibers and transferred to the cold gas stream of an Xcalibur Oxford Diffraction diffractometer or Bruker Apex Duo equipped with low-temperature attachments. Data were collected using monochromated Mo Kα radiation (λ = 0.71073 Å). The scan type was ω. Absorption corrections based on multiple scans were applied with the program SADABS [51], or using spherical harmonics implemented in the SCALE3 ABSPACK scaling algorithm [52]. The structures were solved with the ShelXS structure solution program using direct methods and using Olex2 as the graphical interface [53].

Complex **1** (yield: 92%): ^1^H NMR (300 MHz, DMSO-d_6_): δ 8.39–8.31 (m, 4H, H_Acr,Im_), 8.04–7.99 (m, 2H, H_Acr_), 7.76–7.73 (m, 2H, H_Acr_), 7.59–7.56 (m, 2H, H_Acr_), 4.19 (s, 3H, H_Me_). ^13^C-{^1^H} NMR (75 MHz, DMSO-d_6_): δ 148.5, 140.5, 137.0, 131.6, 129.1, 128.3, 127.9, 126.0, 123.4, 121.9, 117.4, 38.1. HRMS (*m*/*z*): 492.0562 [M-2Cl + H]^+^, 560.98 C_17_H_13_N_3_Cl_3_Au.

Complex **2** (yield: 73%): ^1^H NMR (400 MHz, Acetone-d_6_): δ 8.32 (d, *J*_H-H_, 8.8 Hz, 2H, H_Acr_), 8.19 (d, *J*_H-H_, 2.0 Hz, 1H, H_Im_), 8.05 (d, *J*_H-H_, 2.0 Hz, 1H, H_Im_), 7.86–7.83 (m, 2H, H_Acr_), 7.51 (m, 2H, H_Acr_), 6.99 (d, *J*_H-H_, 8.9 Hz, 2H, H_Acr_), 4.44 (s, 3H, H_Me_). ^13^C-{^1^H} NMR (101 MHz, Acetone-d_6_): δ 150.3, 134.1, 131.6, 130.7, 128.7, 125.4, 124.4, 123.9, 123.7, 38.5. ^19^F-{^1^H} NMR (376 MHz, Acetone-d_6_): δ −123.18 (m), −125.37 (m), −160.08 (t, *J_F-F_* = 19.6 Hz), −161.5 (t, *J_F-F_* = 19.3 Hz), −163.67 (m), −164.70 (m). HRMS (*m*/*z*): 790.0523 [M-Cl]^+^, 825.03 C_29_H_13_N_3_ClF_10_Au.

Complex **3** (yield: 79%): ^1^H NMR (300 MHz, Acetone-d_6_): δ 8.30 (d, *J*_H-H_, 8.8 Hz, 2H, H_Acr_), 8.11 (d, *J*_H-H_, 2.0 Hz, 1H, H_Im_), 8.01 (d, *J*_H-H_, 2.0 Hz, 1H, H_Im_), 7.98–7.87 (m, 2H, H_Acr_), 7.60–7.54 (m, 2H, H_Acr_), 7.37 (d, *J*_H-H_, 8.7 Hz, 2H, H_Acr_), 4.21 (s, 3H, H_Me_). ^13^C-{^1^H} NMR (75 MHz, Acetone-d_6_): δ 149.2, 140.5, 137.9, 133.2, 130.6, 129.7, 127.7, 127.2, 126.0, 122.7, 122.0, 120.9, 117.0, 38.2. ^19^F-{^1^H} NMR (282 MHz, Acetone-d_6_): δ −120.90 (d, 21.12 Hz), −122.58 (m), −159.86 (t, *J_F-F_* = 19.5 Hz), −160.52 (t, *J_F-F_* = 19.4 Hz), −162.97 (t, *J_F-F_* = 19.1 Hz), −164.42 (t, *J_F-F_* = 15.8 Hz). HRMS (*m*/*z*): 958.0623 [M + H]^+^, 957.45 C_35_H_13_N_3_F_15_Au.

1,3-Bis-(2,6-diisopropylphenyl)imidazolinium tetrakis(pentafluorophenyl)aurate(III) (**4**) (yield: 53%): ^1^H NMR (300 MHz, Chloroform-d): δ 7.72 (s, 1H, H_im_), 7.54 (t, J_H-H_, 7.81 Hz, 2H, H_Ph_), 7.32 (d, J_H-H_, 7.82 Hz, 1H, H_Ph_), 4.65 (s, 4H, H_im_), 2.97 (p, J_H-H_, 6.8 Hz, 4H, H_ipr_), 1.38 (d, J_H-H_, 6.8 Hz, 12H, H_Me_), 1.26 (d, J_H-H_, 6.8 Hz, 12H, H_Me_). ^13^C-{^1^H} NMR (75 MHz, Chloroform-d): δ 157.9, 146.1, 132.3, 128.7, 125.4, 54.3, 29.5, 25.4, 23.8. ^19^F-{^1^H} NMR (282 MHz, Chloroform-d): δ −122.38 (m), −159.50 (t, *J_F-F_* = 19.9 Hz), −164.59 (m). ESI^+^ (*m*/*z*): 391.31 [M-Au(C_6_F_5_)_4_]^+^, ESI^−^ (*m*/*z*): 864.9325 [Au(C_6_F_5_)_4_]^−^.

## 4. Conclusions

Several acridine-based gold(III)-NHC complexes were synthesized and characterized by NMR spectroscopy, X-ray diffraction analysis, mass spectrometry and DFT-based calculations. The incorporation of the pentafluorophenyl group (complexes **2** and **3**), confirmed by the ^19^F NMR spectra, improved the solubility of the gold complex in polar solvents with respect to complex **1**. As previously observed for analogue gold(I) complexes, the electronic spectra of the Au(III) derivatives displayed structured bands with maxima at 250 and 360 nm. Based on electronic structure calculations, these bands were associated mostly with π*→π* or σ*→π*. Bond lengths and angles observed for complexes **1** (Trichloro[1-methyl-3-(9-acridine)imidazol-2-ylidene]gold(III)) and **2** (Chloro-bis(pentafluorophenyl)[1-methyl-3-(9-acridine)imidazol-2-ylidene]gold(III)) analyzed by X-ray diffraction were similar to those displayed for other gold(III)-NHC complexes containing pentafluorophenyl groups. As expected, due to the presence of the acridine moiety, both complexes presented π–π stacking interaction in their solid structures. This fact is supported by the calculated interplanar and centroid–centroid distances. The reactivity using a bulkier NHC ligand such as SIPr was also explored. The experimental observations together with computational calculations support the idea that bulky N-heterocyclic carbene ligands, likely leading to higher barriers, can limit the formation of complexes of the form [Au(C_6_F_5_)_3_(NHC)]. This is supported by experimental evidence, and also by the differences observed in the reaction energies for complexes **3**, **4** and **4***.

## Data Availability

Data is contained within the article or Appendix A.

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
