# Peer review of "Steric and Electronic Effects in N-Heterocyclic Carbene Gold(III) Complexes: An Experimental and Computational Study"

_molecules, 2022, doi:10.3390/molecules27238289_

Round 1

Reviewer 1 Report

The synthesis and characterization of novel Au(III) NHC complexes decored with the acridine residue is the main focus of this investigation.

Although I am a computational scientist, and my review posed a higher focus on computational studies (see below), I appreciated both the conceptualization and the structure of the manuscript.  

The information and data are correctly and clearly conveyed to the reader, and the manuscript is well-written.

On the other hand, I found both major and minor issues that the authors must solve to make this manuscript ready for publication on Molecules.   

Majors:

- The computational details are correctly described in SI. My only concern is about the energy of each optimized structure: is that electronic energy? Is that electronic energy plus zero-point correction? Is that electronic energy + zero-point + ideal gas corrections to get H or G at 298.15 K?

  The authors performed the vibrational frequencies analysis for each optimized geometry so they are expected to at least apply the zero-point energy corrections.

- The absolute values of the calculated relative energies of intermediates and products are rather high, almost unrealistic. The authors must provide some explanations/interpretations of that. 

- Therefore, on page 9 lines 250-253, the authors made considerations on the possibly lower activation barrier for the formation of 4. This conclusion might be correct in principle, but it cannot be instead deduced from the calculated data. I suggest to rephrase the mentioned text.

- I do not like very much the rendition of the FO for both complexes and fragments given in Figure 9 and Figure S23-S26. The authors should rather replace these figures with the corresponding correlation diagrams, limited to the carbene-to-metal sigma donation, including the picture of the FOs they wanted to show.

Minors:

- page 4, line 91:  "Since the small size of the methyl group on the imidazolium salt, ..." must be  "Because of the small size of the methyl group on the imidazolium salt, ..."

- page 8, line 209: "... characteristic of the NCHN proton of the N-heterocycle  ..."  is better expressed by  "... characteristic of the N(N)C-H proton of the N-heterocycle  ..."

- page 9, Figure 8: Please magnify the a,b,c labels

- page 9, line 239: I do not like very much the statement "some computational experiments", I suggest to replace it with "calculations"

- page 9, line 251/52: "as well as a less steric hinderance" must be "as well as a lesser steric hinderance"   

- page 10, Figure 9 caption: Please correct "3d" with "5d".    

- page 10, line 270: "... is a n type ..." must be corrected in "... is a p type ..."  

Reviewer 2 Report

Thanks for the opportunity to review the manuscript titled, “Steric and Electronic Effects in N-Heterocyclic Carbene Gold(III) Complexes. An Experimental and Computational Study” by Visbal and co-workers. Understanding the steric and electronic effects in NHC complexes is an important step for the related catalytic reactions and also the development of potential anticancer agents. The current manuscript presented the studies of four [Au(III)-NHC complexes. Reasonable experiments were performed, and interesting results were obtained.

 The following comments need to be addressed prior to publication.

1. Abstract: Please provide the full name of complexes 1 and 2. Please also address it in the conclusion section.

2. Figure 1: Is silver(I) or gold(I) for the metal in the structure presented on the top-right? If so, what is the counterion?

3. Table 2: The energies for the reaction from 3 to Int4 and the reaction from 4 to Int4 are identical. Please explain it.

4. Are the proposed transition state structures presented in Scheme 2 and Scheme 3 computed? Or reported by other literature? It is less likely that the products 3/4 could be formed by only one transition state.

5. The centroid-centroid distances were used to determine the pai-pai stacking interactions in Figures 5 and 6. Please compare the strength of the pai-pai stacking to other NHC analogs?

6. What is the isovalue used for HOMO/LUMO in Figure 9? Why the LUMO of Au-NHC was presented in Figure 9? The authors want to show the orbital contribution from Au in the formation of Au-NHC complex, but there are no electrons at all for LUMO and an orbital node is also presented between Au-C in the Au-NHC complex.

7. The DFT energies and xyz coordinates for optimized structures should be included in the supplementary file. 
